# First Evidence of Microplastic Occurrence in the Marine and Freshwater Environments in a Remote Polar Region of the Kola Peninsula and a Correlation with Human Presence

**DOI:** 10.3390/biology12020259

**Published:** 2023-02-06

**Authors:** Anita Kaliszewicz, Ninel Panteleeva, Kamil Karaban, Tomasz Runka, Michał Winczek, Ewa Beck, Agnieszka Poniatowska, Izabella Olejniczak, Paweł Boniecki, Elena V. Golovanova, Jerzy Romanowski

**Affiliations:** 1Institute of Biological Sciences, Cardinal Stefan Wyszynski University in Warsaw, Wóycickiego 1/3, 01-938 Warsaw, Poland; 2Faculty of Biology and Environmental Sciences, Cardinal Stefan Wyszynski University in Warsaw, Wóycickiego 1/3, 01-938 Warsaw, Poland; 3Faculty of Materials Engineering and Technical Physics, Poznan University of Technology, Piotrowo 3, 60-965 Poznan, Poland

**Keywords:** microfibers, anthropogenic impact, water pollution, rayon, Raman spectroscopy

## Abstract

**Simple Summary:**

The level of microplastic pollution in marine and fresh waters in the least populated and northernmost region of the Kola Peninsula was determined. Fibers were found to be the main components of the microplastics detected and constituted 76–83%. There was a direct anthropogenic effect on the concentration of microplastics in water in the studied lakes in the Kola Peninsula. We established a strong correlation between the number of microfibers and the distance of the lake from the field station. This is the first study that has reported a correlation of microplastic levels with human presence in the most remote lakes in the tundra region in Europe, north of the Arctic Circle of the Kola Peninsula. Our results show that even the occasional human presence can have an impact on microplastic pollution in remote environments.

**Abstract:**

Microplastics (MPs) have even been detected in remote environments, including high-latitude regions, where human activities are restricted or strongly limited. We investigated the surface water of the bays of the Barents Sea and the freshwater lakes that are located close to and several kilometers from a year-round resident field station in the remote tundra region of the Kola Peninsula. The microplastics’ presence in aquatic environments in this region has not been indicated yet. Microplastics were detected in all samples collected from the Barents Sea (<4800 items·m^−3^) and the lakes (<3900 items·m^−3^). Fibers made from polyethylene terephthalate (PET)—the most common thermoplastic polymer of the polyester family—and semi-synthetic cellulosic rayon were the most dominant. This indicated that the source of fiber contamination may come from protective clothes, ropes, ship equipment, and fishing nets. Small microplastics can spread through current and atmospheric transport. The Norwegian Current is likely responsible for the lack of correlations found between MP contamination and the distance from the field station between the studied bays of the Barents Sea. On the contrary, a significant correlation with human presence was observed in the concentration of microfibers in the water of the tundra lakes. The number of MP fibers decreased with an increase in the distance from the field station. This is the first study, to the best of our knowledge, that reports such a correlation in a remote region. We also discuss implications for animals. Our results show that even the most isolated ecosystems are not free from microplastic pollution.

## 1. Introduction

The widespread environmental contamination caused by microplastic (MP) is now well documented and indisputable. MPs, defined as plastic particles that range from 1 µm to 5 mm in size, have been found worldwide—from surface marine waters to deep sea sediment, as well as freshwater systems, ice covers, drinking water, and soil ecosystems [1,2,3,4,5,6]. MP particles have also been detected in aquatic animals or those associated with water, such as mussels, fish, penguins, whales, dolphins, and seals [2,7,8,9]. It seems that no ecosystem is free of MP pollution and that no living organism can avoid its impact. Even remote environments where human activities are restricted or limited have been reported to be polluted by MP debris [10,11,12]. In alpine glacier environments, fibers dominated and represented 65% of MP items. Most of them were made of polyesters [13]. In an alpine region, fibers were detected as the most abundant fraction of MP in an uninhabited lake [14]. MPs were also found in the surface water and sediments of lakes (up to ~1000 items/m^2^) and in the soil of the Tibetan Plateau [15,16]. Fibers were the most frequently observed component of the MP found in surface water and sediment, while the most common form observed in the soil was that of a film. Most fibers that occur in aquatic ecosystems are composed of polyethylene terephthalate (PET) [17,18]. Polyethylene (PE), polystyrene (PS), and PET were the most common polymers detected in the surface water and sediment of the rivers and lakes of the Tibetan Plateau [19,20].

The occurrence of MPs has also been documented for high-latitude environments that are uninhabited or sparsely populated. Gonzalez–Pleiter et al. [21] found MPs in freshwater bodies located in Byers Peninsula, which is an Antarctic Specially Protected Area. Here, mostly polyester and acrylic fibers were detected. Microfibers were also dominant (90% of plastic debris) in samples of sediments obtained from an Arctic freshwater lake located on the island of Spitzbergen of the Svalbard archipelago [22]. The polyester fibers detected in remote polar regions can come from the materials used for manufacturing fleeces and other warm clothing [23]. Fibers constituted the major component of MP found in many polar studies: in the Arctic sea surface, subsurface waters, ice, and snow [24,25,26]. The absence of human settlements and a presence of microplastics in remote environments, including the polar regions, indicate that the most likely cause of MP deposition is atmospheric transport or tourism [21,25,27]. Atmospheric circulation seems to be an important factor for explaining long-range MP movement, and fibers seem to be the components of MP that are easily transported [28]. On the other hand, in polar regions where human activities can be observed (e.g., scientific field stations), MPs can be spread directly by humans.

In this context, we tested whether there was a correlation between the presence of a human settlement in a remote tundra region north of the Arctic Circle and the level of the contamination of the local aquatic environments caused by MPs. Is it proportional to the distance from said human settlement? We examined water samples from the bays of the Barents Sea and from freshwater lakes that are located close to and several kilometers from a year-round residential field station at Dalnye Zelentsy in the remote, scarcely populated region of the Kola Peninsula (69°12′ N; 36°07′ E).

## 2. Materials and Methods

### 2.1. The Study Sites and Sample Collection and Preparation

Eighteen water samples (three replicates per site) were collected in 2019 from six bays of the Barents Sea: two that are located close to the field station in Dalnye Zelentsy (Zelenetskaya: 69°12′ N; 36°07′ E; Yarnyshnaya: 69°12′ N; 36°05′ E), one located 1.4 km away (Avarijnaya: 69°13′ N; 36°07′ E), one located 2.1 km away (Plokhie Chevry: 69°12′ N; 36°12′ E), the fifth located 6.2 km away (Medvezhya: 69°09′ N; 36°22′ E), and the last one located 8.0 km away from the station (Porchnikha: 69°08′ N; 3626′ E). Nine water samples (three replicates per site) were collected from three isolated freshwater lakes located in the tundra at a distance of 0.4 km (69°12′ N; 36°06′ E), 2.1 km (69°10′ N; 36°10′ E), and 5.3 km (69°08′ N; 36°16′ E) from the field station (Figure 1). The field station is inhabited by a few dozen people.

Samples were collected over the course of a week during the polar summer. We used a dense plankton net (20 µm) to also detect small MPs. The plankton net was flushed three times before sampling. The net with an inlet that had a diameter of 23 cm was trawled just below the water surface on a transect of 2 m, and at a distance of about 5 m from the shore. For each sample, 14.4 L of water was passed through the plankton net to obtain a volume of about 50 mL. Care was taken to avoid contamination from the person towing the plankton net. There was a low probability of contamination of the field samples with fibers from the net. Its fibers were transparent and oval in shape and 34.5 µm wide, compared to 10–24 µm wide and mostly different in color and ribbon-shaped microfibers from the field samples. The samples were placed in 100 mL containers that had screw caps. After being transported to the laboratory, the samples were transferred into glass flasks and dried at 60 °C. According to the widely used methodology modified by us [29,30], the dried samples were immersed in a 30% hydrogen peroxide and 69% nitric acid solution by a proportion of 3:1. The flasks were then covered with glass plates and heated to 70 °C on a magnetic stirrer (Ika) for 72 h for organic matter digestion. Next, the samples were filtered using the vacuum pump kit Labor s. c. PL2/1 SN 1309 with glass microfiber filters that were 47 mm in diameter and had a pore size of 1.2 µm (Whatman, GF/C^TM^). The filters were placed individually in glass Petri dishes with a lid and left to dry for 24 h. Each filter was visually analyzed under the stereoscopic microscope Huvitz HSZ-ZB700, and MPs were photographed using a Keyence VHX-7000 digital microscope at 500–1000× magnification. The MP particles were manually counted and individually measured from the images using the software of the Keyence VHX-7000 digital microscope.

### 2.2. Negative Control Samples

Negative control is important to assess potential contamination of samples by the collection and extraction procedures. To check whether there were any MP particles resulting from the sampling method, we poured 14.4 L of deionized water through the plankton net (three replicates) and treated with similar procedure used for the samples from the study sites: dried, filtered, placed in Petri dishes, visually analyzed, and possible MPs photographed using a Keyence VHX-7000 digital microscope at 500–1000× magnification.

We also checked whether there were any MP particles in the laboratory air. We used clean glass microfiber filters (three replicates) that were placed in the opened Petri dishes for 4 h in the working area of the laboratory. Each filter was visually analyzed under the stereoscopic microscope Huvitz HSZ-ZB700, and possible MPs were photographed using a Keyence VHX-7000 digital microscope at 500–1000x magnification.

### 2.3. Identification of MPs

We analyzed the representative fibers using the Raman spectroscopy. This method has been successfully used to identify MPs in the environment [31,32,33]. The Raman spectra of MP and natural fibers were recorded using a Renishaw inVia Raman spectrometer that was equipped with a thermoelectrically (TE) cooled CCD detector, which is a semiconductor laser-emitting light in the near-infrared region of 785 nm wavelength. The spectra were recorded in the spectral range of 100–3200 cm^−1^ with a spectral resolution of better than 2 cm^−1^. The power of the laser beam that was focused on the sample with a 100× objective was kept below 10 mW. The position of the peaks was calibrated using a crystalline Si before the data were collected. To identify plastic type, the spectra were compared to a spectral database of commonly known polymers. The spectral parameters of the bands, such as peak center position, intensity, integral intensity, and FWHM (full width at half maximum), were determined using the fitting procedure provided in the Wire 3.1 software.

### 2.4. Statistical Analyses

The data were analyzed for normality using the Shapiro–Wilk test. The differences in the MP volume and the proportion of particular components across study sites were analyzed using a one-way ANOVA test, followed by a post-hoc Tukey’s test. Pearson’s correlation was used to determine the relationship between the volume of MPs and the distance from the field station. A significance level of α = 0.05 was used for the statistical analysis. All statistical analyses were performed using Statistica 12 (StatSoft, Inc., Tulsa, OK, USA).

## 3. Results

### 3.1. MP Debris in the Surface Water of the Barents Sea

MP fibers and fragments were found in all the water samples collected from the studied bays. The contamination detected was in the range of 1500–4170 fibers·m^−3^ and 420–1600 fragments·m^−3^ (Figure 2). Fibers constituted the most abundant type of MP pollution (76 ± 4%).

There were significant differences between the number of fibers and fragments observed for two bays located close to the field station: Zelenetskaya and Yarnyshnaya (Tukey’s test, *p* = 0.04 and *p* = 0.008, respectively). The highest number of fibers was observed in Yarnyshnaya, and the lowest was seen in Plokhie Chevry; however, there were no statistically significant differences between the studied bays (ANOVA, *p* > 0.05). The number of fragments also did not differ between these sites (ANOVA, *p* > 0.05).

There were no significant correlations found between MP contamination (number of fibers and fragments) and the distance from the field station (Figure 2).

The fibers and fragments were divided into four size classes: (a) small (0.01–0.20 mm), (b) medium (0.21–1.00 mm), (c) large (1.01–5.00 mm), and (d) mesoplastic (5.01–25.00 mm). Fibers of all size classes were detected (Figure 3a). The proportion of medium and large fibers was the highest (49 ± 9% and 34 ± 7% of all fibers, respectively), while that of the small fibers differed significantly between two bays: Porchnikha, which is the farthest from the field station, and Plokhie Chevry, which is located 2.1 km away (Tukey’s test, *p* = 0.04). The mesoplastic fibers were less abundant, and accounted for ≤2% of all the fibers detected (Figure 3A).

Among fragments, the small ones were the most prevalent (97 ± 5% of all fragments detected were in the range of 0.01–0.20 mm), while 0–6% were 0.21–1.00 mm in size. There were no mesofragments (5.01–25.00 mm) found (Figure 3b).

Large-sized and mesoplastic fragments were not detected. There were no significant differences in the proportions of fragments with regard to particular size classes between and within the studied sites (ANOVA, *p* > 0.05).

### 3.2. MP Contamination in the Surface Water of Lakes: Significant Correlation with Human Activity

The scale of MP contamination and the proportion of fibers found in the lakes were similar to those observed in the bays: 1100–3300 fibers·m^−3^ (Figure 4A). The number of fragments was 185–600 particles·m^−3^, which is fewer than that in the bays, but the difference was not statistically significant (ANOVA, *p* > 0.05). Fibers constituted the most abundant type of MP pollution (83 ± 2%). There were significant differences between the number of fibers and fragments within Lake 1 (located close to the field station) and Lake 3 (farthest from the field station) (Tukey’s test, *p* = 0.0003 and *p* = 0.01, respectively; Figure 4A).

When we compared the lakes themselves, we noted a significant negative correlation between the number of fibers and the distance from the field station (Pearson’s correlation, r = −0.73, *p* = 0.02; Figure 4B). The number of MP fibers decreased with an increase in the distance from the field station. No such significant correlation was found for MP fragments despite the fact that the fewest number of fragments were recorded in Lake 3, which is farthest from the station (Figure 4A).

As was performed for the bays, the fibers and fragments found in the lakes were divided into four size classes: (a) small (0.01–0.20 mm), (b) medium (0.21–1.00 mm), (c) large (1.01–5.00 mm), and (d) mesoplastic (5.01–25.00 mm). Fibers of all size classes were detected only in Lake 1, which is the closest to the field station; in Lake 2 and 3, the mesoplastic fibers were not found (Figure 5a). The mesoplastic fibers (5.01–25.00 mm) found in Lake 1 accounted for ≤2.4% of all the fibers detected. Medium fibers were dominant in all lakes (55 ± 6% of all fibers). The proportion of the small fibers was lower than that of the medium ones (21 ± 10%), and differed significantly between lakes (ANOVA, F_2,6_ = 11.1, *p* = 0.01). The percentage of the small fibers was the lowest in Lake 3, as compared to Lake 1 and 2 (Tukey’s test, *p* = 0.002 and *p* = 0.01, respectively; Figure 5a).

As for MP fragments, similar to the situation in the bays, the small ones were dominant (93 ± 10% of all fragments detected were in the range of 0.01–0.20 mm; Figure 5b), and large fragments were not detected. There were significant differences in the proportions of medium (0.21–1.00 mm) fragments across the lakes (ANOVA, F_2,6_ = 17.6, *p* = 0.003). The highest percentage of medium fibers was found in Lake 1, as compared to Lake 2 and 3 (Tukey’s test, *p* = 0.01 and *p* = 0.003, respectively; Figure 5b). Lake 1 was the only lake in which large fragments were detected. The results for Lake 3, which is farthest from the field station, indicated the presence of only small fragments (Figure 5b).

### 3.3. Method-Driven Contamination–Negative Control

Microplastic contamination resulting from the sampling method may constitute 12%. We found only the fibers. The medium fibers dominated and constituted 62% of all detected in the control test. The mesoplastic fibers were not present. There were no microplastic contaminants found on the filters during air exposure.

### 3.4. Polymers Identified

We found similar MP polymers in the Barents Sea and in the freshwater lakes, and PET was the most common among them (Figure 6).

On average, 67% of the fibers found in the bays and 60% of those found in the lakes made from this polymer. The PET fibers were the most abundant (100%) in Lake 1, which is closest to the field station, and Barents Sea, but the results were not statistically significant (Figure 4A). The remaining fibers (on average, 33% of fibers in the bays and 40% in the lakes) were identified as rayon, which are semi-synthetic cellulose-based fibers (Figure 7).

The PET and rayon fibers can come from the equipment used on ships, fishing nets, and protective clothing. These source materials were all observed in and around the field station. The diagram shows the possible pathways for plastic fragmentation and spread in this region (Figure 8).

## 4. Discussion

### 4.1. Anthropogenic Effect on the Microplastic Concentration

MPs were detected in the studied surface water samples from the Barents Sea and the lakes in the remote region of the Kola Peninsula. To the best of our knowledge, there have been no previous studies on MP contamination in this area. MP contamination has already been reported in the other polar regions [21,34]. We found higher MP abundance in the waters of the Kola Peninsula than that reported for the Arctic polar waters (by one order of magnitude [35]; or two orders of magnitude [5,36]), and a similar number as reported for the sediments from Arctic and Tibetan lakes and the water from remote alpine lakes [14,15,22]. On the other hand, the concentration of MP in the waters of the Kola Peninsula reported in this study was one order of magnitude smaller than that in seawater samples taken from the western Antarctic Peninsula [37]. The differences observed in the level of microplastic contamination between studies may emerge not only from the MP distribution, but also the methods used, especially the mesh size of the net used. In many studies, nets with a mesh of 125–330 µm have been used. In the current study, we used a dense net with a mesh of 20 µm. Finer filtering mesh led to a larger number and smaller size of microplastics [38].

The direct anthropogenic effect on the concentration of MPs in water was indicated for the studied lakes in the Kola Peninsula. We established a strong correlation between the number of microfibers and the distance of the lake from the field station inhabited for an entire year. To the best of our knowledge, this is the first paper to indicate such a direct relationship between the MP contamination level in freshwaters and human presence in a remote region of Kola Peninsula. Previous studies have reported increases in MP pollution in marine environments near populated areas [39,40]. MP abundance was higher in the sediments obtained from the urban site than those from the rural site of the southern part of South Korea. Dowarah and Devipriya [41] observed a correlation between fishing activity and MP abundance in the sediments of beaches on the coast of Puducherry, India. Despite the assumption that such a relationship between the abundance of MP and human activity should exist in inland waters, this has not been demonstrated until now. For instance, Yuan et al. [42] did not find a correlation between eutrophication, considered an anthropogenic process, and MP pollution in the freshwater Dianchi Lake in China. The difficulty associated with demonstrating such a correlation may be due to the transport of MPs by water currents and wind. These ways of microplastic distribution, especially fibers, seem to play an important role in increasing pollution [43]. Fibers are the most prevalent component of the MPs found in the air [23,28]. The significance of long-range transport has been also documented for remote and polar regions [27,34]. The Norwegian Current is likely responsible for the lack of correlations found between MP contamination and the distance from the field station to the studied bays of the Barents Sea. The potential pathways of fragmentation of plastic and the transport vectors of microplastics to aquatic environments in the studied region of the Kola Peninsula are presented in a diagram (Figure 8). The presence of microfibers in the studied lakes located 2–5 km away from the field station in the Kola Peninsula could mainly be explained by wind transport. There are no other inhabited places within a radius of several dozen kilometers, and this area is characterized by a windy climate resulting from the main North Atlantic storm track [44,45]. The low levels of tourism and car traffic seem to have less impact on fiber transport than wind. Detailed studies on the MP present in the air could confirm this assumption.

A strong correlation between the number of microfibers and the distance of the lake to the field station indicated its main role in distributing MPs. A proposition to reduce the microplastic contamination originating from the field station and to protect the water and tundra ecosystems includes (1) increasing environmental awareness among residents and station users to sort waste; (2) reorganizing the landfill, storing plastic waste in a separate pile, and making it easier to secure it; (3) regularly securing waste at the landfill, e.g., by covering the pile with a layer of soil/sand; and (4) suggesting that visitors to the station take plastic waste with them to the city, where there are facilities to process it.

### 4.2. PET and Rayon Microfibers Dominate in Arctic Waters

Fibers were found to be the main components of the MPs detected in the surface water of the Barents Sea and the lakes included in this study. Further, fibers have been reported as the dominant component found in water and sediment samples in many previous studies on remote and polar regions [15,22,25]. Moreover, fibers also constitute a major type of MP in marine animals, e.g., benthic organisms from the Arctic and sub-Arctic regions [46]. Desforges et al. [36] have found that, in the subsurface waters of the northeastern Pacific Ocean, nearshore samples presented more fiber content than offshore samples. Fibers can originate from a variety of sources, including textiles, synthetic fishing nets, and ropes [32]. Up to 60% of the textiles produced today are synthetic, with the main polymers used being polyester, which includes PET (the most common thermoplastic polymer of the polyester family), acrylic, and nylon [47]. In our samples, mostly, PET fibers were detected. Regarding production volume, 70% of synthetic fibers are made of PET [48]. Fibers made from PET are used to produce knitted and woven fabrics, including fleece, Dacron, tergal (e.g., for sail cloth), and rope. Their presence in the studied samples from the Barents Sea and the lakes in the tundra region beyond the Arctic Circle is, therefore, not coincidental. They can come from the equipment used on ships, fishing nets, and protective clothing. A large number of previous studies have indicated that PET fibers are pervasive (up to 94%) in aquatic environments, including marine and lake water columns and sediments [49,50,51,52].

The second most common component of the fibers detected in our study was rayon, which is a semi-synthetic cellulosic material. The main source of rayon can be assumed to be the clothing from which it is released. Rayon has been reported in many polar environments as an abundant component of the microfibers detected in water, sediments, and sea ice [24,25,53]. The rayon fibers could have been labeled as being less hazardous to the environment than synthetic polymers if they had not been the most frequently detected MP in organisms [54]. However, rayon was found in guts of fish [55,56] and reported as the most common fiber (53%) detected in the True’s beaked whale [57]. Even natural fibers may contain additives, such as dyes, antioxidants, plasticizers, resins, and flame retardants, which pose risks for the organisms that ingest them [58].

The microfibers composed of PET or rayon found in this study in water samples from the remote region of the Kola Peninsula might come from clothing in the course of normal wear and through atmospheric transport. This could be especially true for the studied tundra lakes, for which a significant correlation between the volume of microfibers and the distance from the field station was established.

### 4.3. Potential Risks to Organisms from Microplastic Pollution–Implications for Polar Animals

The Barents Sea is one of the most productive seas in the world. The primary production in the water column of this sea has increased over a 65-year long period (average annual primary production of 103 g C m^−2^ y^−1^), and is grazed by the dominating copepods, *Calanus finmarchicus* and *Calanus glacialis* (the average zooplankton biomass is 3.5 g C m^−2^ y^−1^) [59]. Zooplankton species are the basis for the rich assemblage of higher trophic level organisms: shrimps, commercial fish stocks such as cod (*Gadus morhua*), capelin (*Mallotus villosus)* and herring (*Clupea harengus)*, sea birds, seals, and whales [60]. The benthic community is dominated by echinoid *Strongylocentrotus droebachiensis*; bivalves *Chlamys islandica* and *Mysella dawsoni*; the bryozoans, *Eucratea loricata* and *Alcyonidium gelatinosum;* the sea cucumber, *Cucumaria frondosa*; the hydroid, *Sertularia mirabilis;* and the brittlestar, *Ophiura robusta.* The coarse sediments are dominated by suspension-feeding animals: cnidarians, bryozoans, sea cucumbers and mollusks, including *Hiatella arctica*, *Macoma calcarea,* and *Ennucula tenuis*, and surface predators, including nemerteans and polynoid polychaetes [61]. Atlantic sea scallops (*Placopecten magellanicus*), sea urchin *Strongylocentrotus droebachiensis*, bivalve *Astarte* spp., and *P. magellanicus* dominated production. In deep parts of the Barents Sea, benthic biomass can reach on average of about 100 g m^−2^ wet weight [62]. Maximum epifaunal production is estimated at 21 g C m^−2^·year^−1^ [61]. Microplastic pollution in the highly productive Barents Sea poses a threat to animals at all levels of the food chain, including humans.

Previous studies have indicated that microplastic particles tend to be directly consumed by animals, such as crabs, cladocerans, and mussels that cannot distinguish it from their food [63]. Microplastics can also stick to the external organs of animals, e.g., gills, or to algae, and can be then consumed by herbivorous invertebrates or fish. Microplastics have been found in the intestines, muscles, and bloodstream of fish. Boerger [64] has shown that 35% of fish caught in the North Pacific have between one and 83 MP particles in their intestines. Microplastic is similar to plankton in size and can be ingested either intentionally or by simple confusion with food, as recorded for fish [65] and filter-feeding animals [66]. For filter-feeding marine species, MPs overlap in size range with prey and selection against inert particles is little developed [31]. The mistaken uptake of MP particles as food has been noticed for invertebrates [67] and within the planktonic food web [68]. It is assumed to be an important factor for transfer between trophic levels. Although reports for polar species are still rare, evidence has emerged regarding large animals, e.g., fish [69,70] and invertebrates such as krill [71].

The consequences of plastic ingestion for such parameters as growth, survival, performance, and reproduction are unknown for Arctic species, although a number of studies have recently investigated the effects [1,72]. Particles of plastic may be retained in the digestive system, simply causing a decrease in feelings of hunger and subsequent reduced intake of food [73]. Nano- and microplastics can enter the blood system and accumulate in internal organs, directly causing inflammation, oxidative damage, and necrosis [74]. There is evidence that the size of the particles limits their translocation: 0.5 mm particles were proven to be translocated to the hemolymph of crabs, *Carcinus maenas* [67], whereas larger particles (8–10 mm) did not appear in its circulatory system.

Macro- and microplastics are available to many marine organisms, ranging from plankton to whales, and may accumulate within individual organisms over time and be transferred up the food web (biomagnification). Farrell and Nelson [67] confirmed the trophic transfer of MP (polystyrene of 0.5 mm in size) from mussels (*Mytilus edulis*) to crabs (*Carcinus maenas*) in a laboratory study. It is likely that the type and number of ingested MPs can vary depending on the species’ foraging strategy: planktivores and filter-feeders are at a higher risk of ingesting low-density plastic particles that float at the sea surface, while deposit feeders are more susceptible to ingesting high-density plastic fragments from sediments, e.g., polyvinyl chloride (PVC) [75,76]. Filter-, suspension- and deposit-feeders are mostly non-selective feeders. Although some species such as holothurians or bivalves feed selectively, they can accidentally consume additional inorganic material, including plastic particles. Predators choose their prey intentionally; thus, ingestion of MP is mostly due to bioaccumulation [77]. Microplastic ingestion has been documented for many invertebrate predators and scavengers, such as the littoral crab, *Carcinus maenas* [67,78]; the European brown shrimp, *Crangon crangon* [31,79]; and the Norway lobster, *Nephrops norvegicus* [77].

Numerous studies have classified species as indicators for MP ingestion; however, there is often no clear rationale for their selection as an indicator species. Species at lower trophic levels can be adequate indicators in monitoring due to their basic position in the food chain [80]. Bivalves and crustaceans, which are consumed whole, are of particular interest for assessing human health risks, as the most common studies have reported MP in the digestive tract of these organisms [81]. Van Cauwenberghe and Janssen [2] reported the presence of MPs (an average of 0.36 ± 0.07 particles g^−1^) in the common mussel, *Mytilus edulis*.

Ingestion rates measured on natural zooplankton communities revealed that 83% of Norway lobster *(Nephrops norvegicus)*, 63% of shrimp *Crangon crangon*, 3% of the copepod *Neocalanus cristatus,* and 6% of *Euphasia pacifica* consumed MPs, mostly fragments or fibers [36,77,79]. Similarly, the data for polychaete *Hediste diversicolor* showed that 58.58% of individuals ingested MP particles, with fibers accounting for 86.8% of the ingested particles [82]. Studies on fish reported that up to 40% were contaminated, a mean number of MP particles from 1 to 7.2 per individual [83]. These results show how much of a threat MP pose to marine animals and what a necessity it is to examine the scale of this biomagnification in marine invertebrates, including those inhabiting the eulittoral of the Arctic-Boreal region.

The detected MP contamination could have implications for freshwater invertebrates and ichthyofauna of tundra lakes in the studied region, e.g., Nematoda, Rotifera, Cladocera, Copepoda, diptera larvae, especially Chironomidae and the fish often observed at these latitudes, trout, and grayling [84,85]. Previous studies have indicated that MP can enter food webs in different freshwater environments and can be ingested by feeding on microplastics-contaminated food [30,86]. Luoto et al. [87] suggested a high probability for Arctic freshwater biota and animals to encounter MPs. However, studies on MP debris in polar freshwater animals are very scarce. The effect of MP on zoobenthos has been described for deposit-feeders chironomidae (no abundance changes observed) and collector-filterers (a decrease in abundance). Zooplankton has been negatively affected by MP fragment consumption, and fine-mesh, filter-feeding cladoceran *Chydorus sphaericus* almost disappeared from waters polluted by MP [88]. Microplastic has been detected in the stomach and intestine of Arctic fish salmonids, and can have negative effects [89]. Published information on MP presence in Arctic fish is scarce and restricted to gut content. The microplastic contamination of freshwater systems is less studied than marine systems, with 96 studies focused on marine organisms and only 21 on freshwater [88]. Polar freshwater animals are still poorly tested for MP contamination and there is a need to extend research to high-latitude, unique freshwater ecosystems in order to assess the level of MP transfer that occurs through trophic levels.

Plastic fragments are also able to carry a broad range of toxic chemicals that increase exposure risk for ingesting organisms [58]. Arctic biota exposed to plastic pollution may be especially vulnerable to contamination from ingested MP, as they are already under environmental stress from climate warming, pollution, and ocean acidification. As a direct consequence of ingested MPs, pollutants can be transferred into the organism’s tissues, causing potential toxicological effects. How these deleterious effects impact Arctic biota, in particular, has not been studied so far, but polar animals’ fat-rich tissues may make them particularly prone to toxicological effects. For this reason, testing the level of MP contamination of organisms from polar regions and assessing the level of MP transfer through trophic levels is very important, not only for these high-latitude, unique ecosystems, but also for humans.

## 5. Conclusions

The surface water of the Barents Sea and freshwater lakes of the remote region of the Kola Peninsula were found contaminated with MPs. The concentration of MPs in the water of the tundra lakes were similar to those observed in the bays, and varied from 1300 to 4800 items·m^−3^. The main components of the MPs detected in the surface water were fibers. They were mostly identified as PET—the most common thermoplastic polymer of the polyester family—or semi-synthetic, cellulosic rayon regardless of the sampling location. A strong correlation was observed between the number of fibers in the lakes and the level of human presence in the remote region. To the best of our knowledge, this is the first paper to indicate such a direct effect. The microfibers found in the lakes might mainly be released from landfill, protective clothing, and transported by the wind. For the studied bays of the Barents Sea, the influx of contaminants due to the Norwegian Current come into play. Research conducted in remote and isolated areas allow an estimate of the global level of microplastic pollution and can provide a baseline of populated areas. These areas, due to their low direct participation of inhabitants, can be considered as indicator areas to assess the current level of pollution. For a better understanding of this global problem, the concentration of microplastic in the environment, as one of the leading environmental problems, should be regularly monitored along with other pollutants.

## Figures and Tables

**Figure 1 biology-12-00259-f001:**
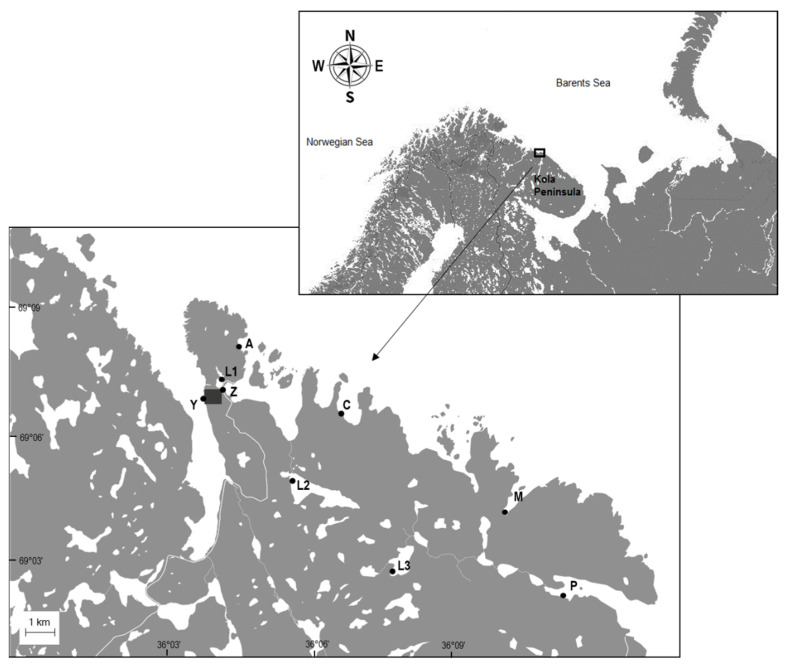
Map of the sampling sites in the Kola Peninsula. Water samples were collected from six bays of the Barents Sea—(A) Avarijnaya, (C) Plokhie Chevry, (M) Medvezhya, (P) Porchnikha, (Y) Yarnyshnaya, and (Z) Zelenetskaya—and three freshwater lakes located in the tundra region—L1, L2, and L3. The field station is marked by a grey square.

**Figure 2 biology-12-00259-f002:**
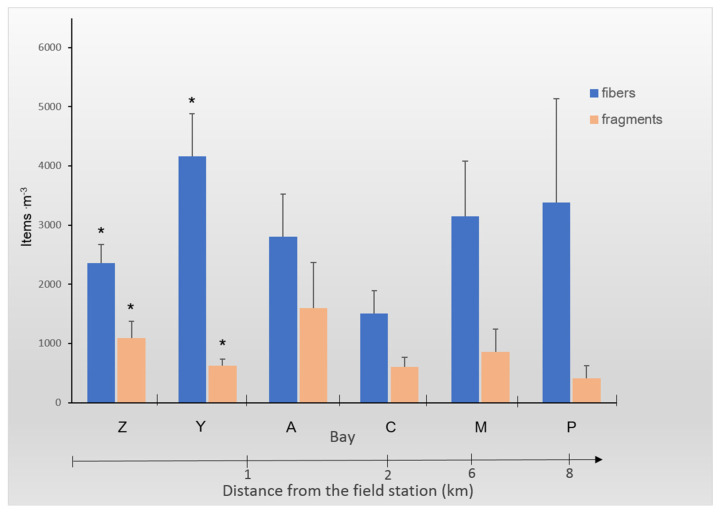
The mean number (particles·m^−3^) of the fibers and fragments found in the surface water of the studied bays of the Barents Sea: (A) Avarijnaya, (C) Plokhie Chevry, (M) Medvezhya, (P) Porchnikha, (Y) Yarnyshnaya, and (Z) Zelenetskaya. The distance (in km) from the field station is shown by the arrow at the bottom of the graph. The error bars represent ± 1SE. Statistically significant differences are indicated by * *p* < 0.05.

**Figure 3 biology-12-00259-f003:**
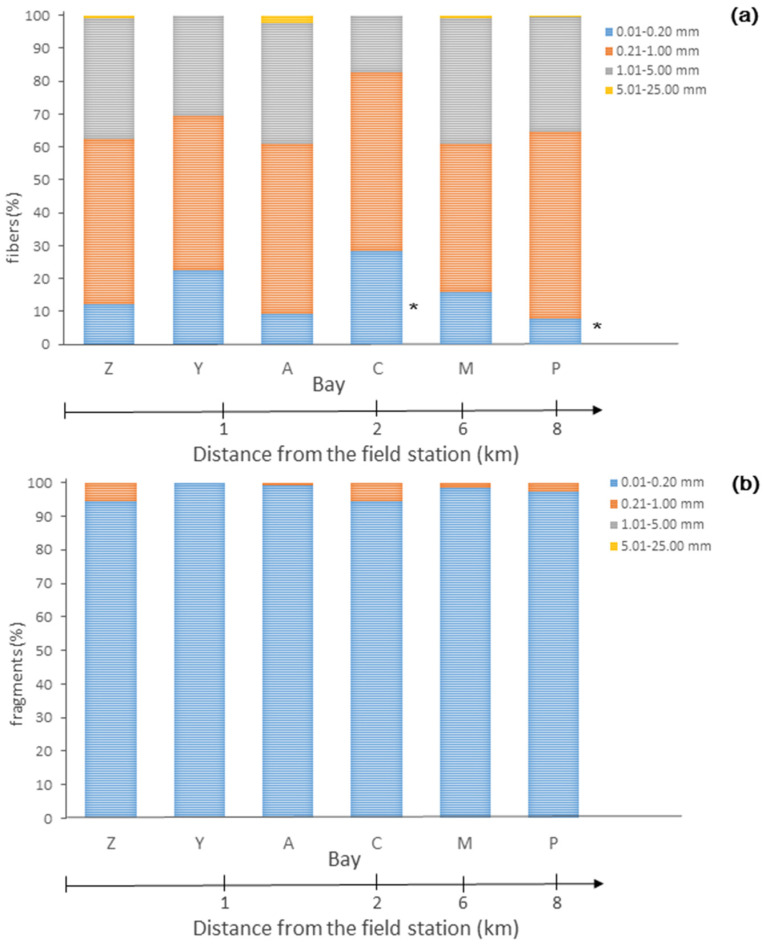
The percentage of microplastic (**a**) fibers and (**b**) fragments of different size classes (0.01–0.20 mm, 0.21–1.00 mm, 1.01–5.00 mm), and mesoplastic (5.01–25.00 mm) in the surface water of the studied bays of the Barents Sea: (A) Avarijnaya, (C) Plokhie Chevry, (M) Medvezhya, (P) Porchnikha, (Y) Yarnyshnaya, and (Z) Zelenetskaya. The distance (in km) from the field station is shown by the arrows at the bottom of the graphs. Statistically significant differences are indicated by * *p* < 0.05.

**Figure 4 biology-12-00259-f004:**
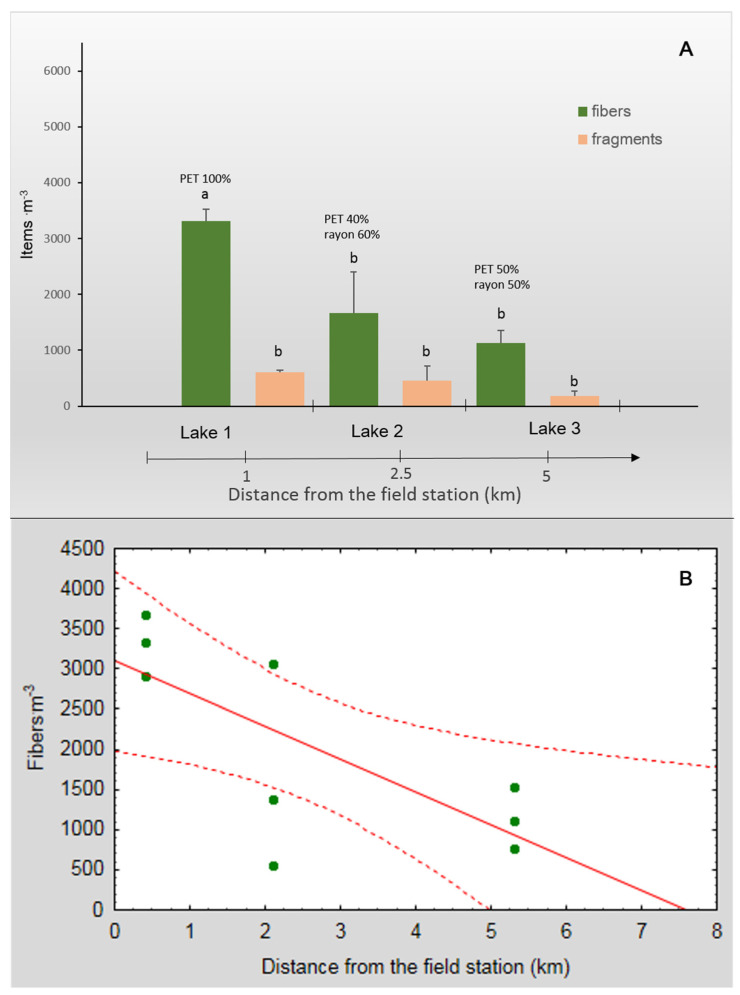
(**A**) The mean number (particles m^−3^) of fibers and fragments found in the surface water of the studied tundra lakes and (**B**) a correlation between the number of fibers and the distance from the field station. The percentages of the different types of fibers are shown: PET and semi-synthetic cellulosic rayon. The bars that share the same letter are not significantly different. Dotted lines represent confidence intervals. The error bars represent ±1SE.

**Figure 5 biology-12-00259-f005:**
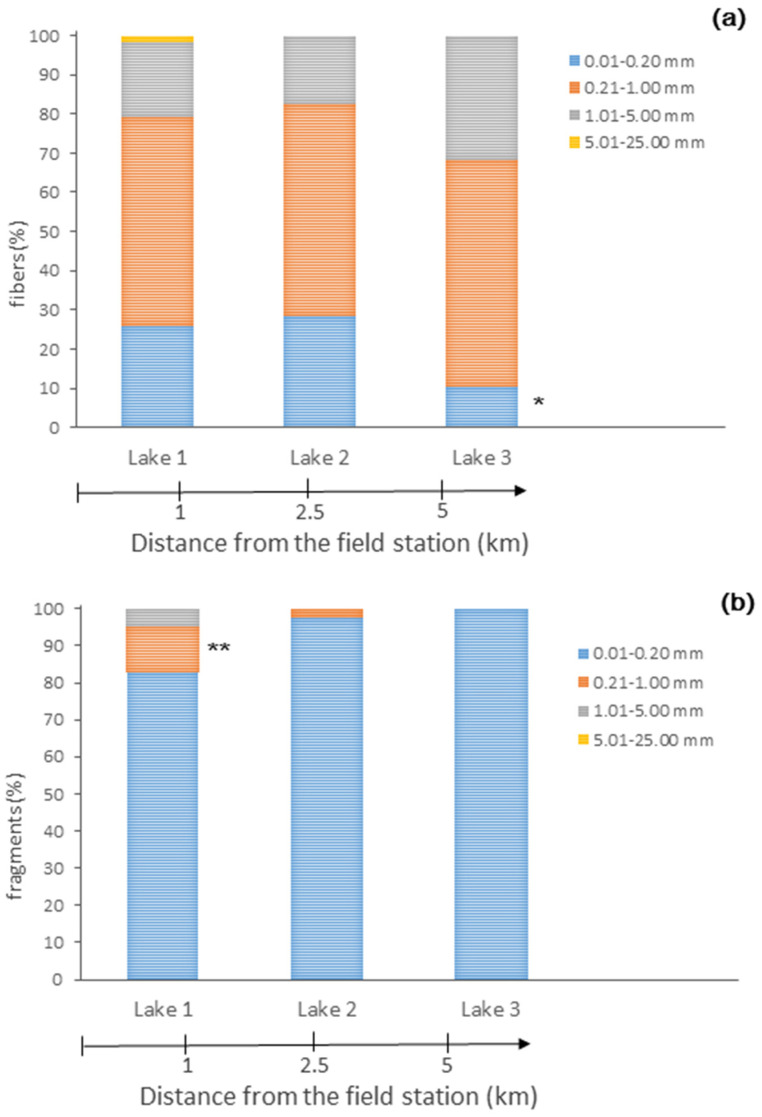
The percentage of microplastic (**a**) fibers, (**b**) fragments of different size classes (0.01–0.20 mm, 0.21–1.00 mm, 1.01–5.00 mm), and mesoplastic (5.01–25 mm) from the surface water of the studied tundra lakes. The distance (in km) from the field station is shown by the arrows at the bottom of the graphs. Statistically significant differences are indicated by * *p* < 0.05 and ** *p* < 0.005.

**Figure 6 biology-12-00259-f006:**
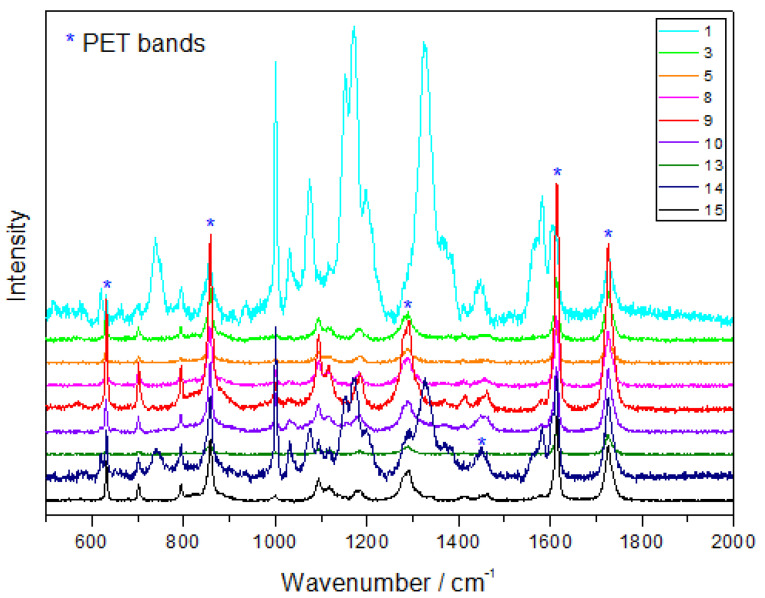
The Raman spectrum of PET, acquired using the Raman spectroscopy. The samples from the Barents Sea are numbered from 1 to 8, and those from the tundra lakes are numbered from 9 to 15.

**Figure 7 biology-12-00259-f007:**
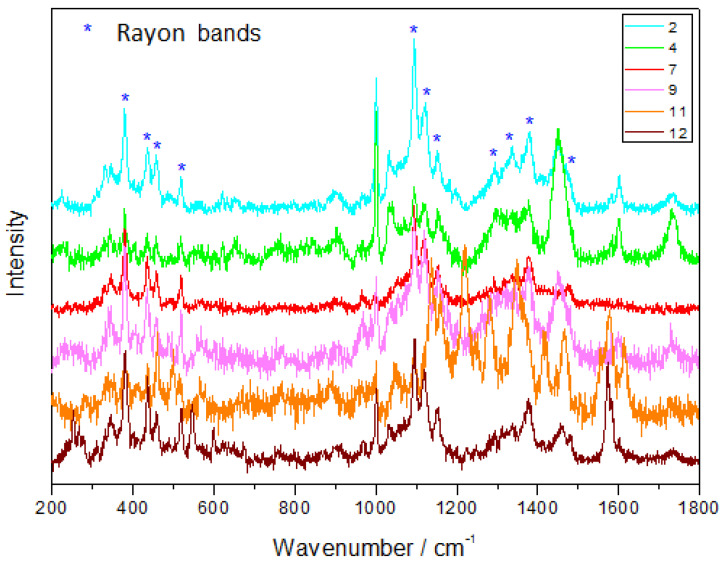
The Raman spectrum of semi-synthetic cellulosic rayon, acquired using the Raman spectroscopy. The samples from the Barents Sea are numbered from 2 to 4, and those from the tundra lakes are numbered from 7 to 12.

**Figure 8 biology-12-00259-f008:**
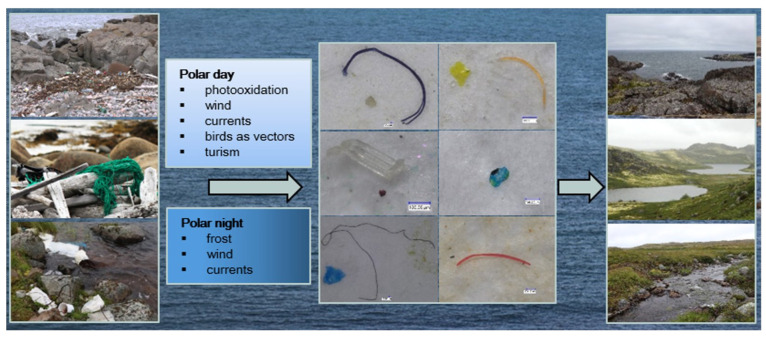
Photographic diagram depicting the main possible ways of fragmentation of plastic (textiles, fishing nets, ropes, pipes, single use packaging, bottles, containers) found in the seashore and tundra and the processes responsible for transport of microplastics (fragments and fibers) to aquatic environments in the Kola Peninsula region north of the Arctic Circle.

## Data Availability

The data presented in this study are available in https://doi.org/10.6084/m9.figshare.22015634.v1.

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
