# Peer review of "First Evidence of Microplastic Occurrence in the Marine and Freshwater Environments in a Remote Polar Region of the Kola Peninsula and a Correlation with Human Presence"

_biology, 2023, doi:10.3390/biology12020259_

Round 1

Reviewer 1 Report (Previous Reviewer 1)

In methodology mention temperature in line no. 116 because microplastic also degrade with high heat.

Size of the particles are not mentioned it is required.

Microsize particles are not discovered. What was the reason.

Only Raman Analysis confirming type of plastic. Pyro-GC MS not done why?  

Other similar work is also done in various part it is not not cited

Such as

1.      Manzoor S, Kaur H, Singh R. Existence of Microplastic as Pollutant in Harike Wetland: An Analysis of Plastic Composition and First Report on Ramsar Wetland of India. Curr World Environ 2021;16(1). Available From : https://bit.ly/3sCt9TI

In discussion Kindly mention about mitigation possibilities.

Author Response

Response to the review in the attached file.

Reviewer 2 Report (New Reviewer)

This manuscript investigated the surface water of the bays of the Barents Sea and the freshwater lakes that are located close to and several kilometers from a year-round resident field station in the remote tundra 30 region of the Kola Peninsula. The manuscript was suggested to be published after major revisions.

 1 The abstract should be rewritten.

2 Containers without plastics should be employed to obtain the MPs samples. However, in this study, plankton net was used. The authors should discuss the effect of plankton net on the MPs concentration of the samples.

3 For each sampling site, how many samples were collected? The detailed information of sample collection should be added.

4 The authors should check carefully throughout the manuscript before submission. There are too many detail problems, e.g., “Figure 4. Fig. 4. (a)”.

5 The format of the references should be checked thoroughly.

Author Response

Response to the review in the attached file.

Reviewer 3 Report (New Reviewer)

General comment

In the manuscript biology-2163361 entitled “First evidence of microplastic occurrence in the marine and freshwater environments in a remote polar region of the Kola Peninsula and a correlation with human presence” Kaliszewicz and co-authors investigated samples of surface water of the bays of the Barents Sea and the freshwater lakes that are located closed to and several kilometers from a year-round resident field station in the remote tundra region of the Kola Peninsula. The study was well conducted and reported important monitoring data to share with the scientific community. M&M are described with enough detail to allow others to replicate and build on published results. The paper provides a concise and precise description of the results. The discussion is well written. Thus, the manuscript deserves to be published in Biology.

While I enjoyed the flow of the paper, I could not overcome the sense that there are some minor issues that could addressed to improve the quality of the manuscript prior to publication.

Specific comments

-       Line 54. I suggest to add some references at the end of the sentence. You should include this study on microplastic occurrence in frogs from high-mountain ponds (https://doi.org/10.3390/d14020066)

-       Line 89. I suggest to add more information about water samples. How many samples per site? How many replicates? Please, add.

-       Section 2.2. needs to be improved to clarify the quality criteria for analyzing MPs in samples. I strongly recommend that authors detail procedures regarding "sampling methods", "sample size", "sample processing and storage", "laboratory preparation", "clean air conditions", "negative controls", "sample treatment", and "polymer identification". 

-       Results of ANOVA test should be also reported in the figure 2 (asterisks for example).

Author Response

Response to the review in the attached file.

Round 2

Reviewer 2 Report (New Reviewer)

Conclusion is loosely written and should be improved precisely. Please suggest significance and prospect for future studies in the conclusion.

Author Response

We would like to thank the reviewer for the constructive and competent criticism. According to the suggestions we rebuilt the structure of the conclusions and added the prospect for future studies (page 19).

This manuscript is a resubmission of an earlier submission. The following is a list of the peer review reports and author responses from that submission.

Round 1

Reviewer 1 Report

It is good work, but there scope to improve quality of work. 

1. Reference of methodology particularly digestion of organic substance is not given. it must be given.  you can cite - Existence of Microplastic as Pollutant in Harike Wetland: An Analysis of Plastic Composition and First Report on Ramsar Wetland of India. (http://dx.doi.org/10.12944/CWE.16.1.12)

2. Why not density separation done for this work. You can refer -Analysis Of Nylon 6 As Microplastic In Harike Wetland By Comparing Its IR Spectra With Virgin Nylon 6 And 6.6 S MANZOOR, H KAUR, R SINGH - European Journal of Molecular & Clinical Medicine, 2020.

3. Timing of digestion is not properly address.

4. What was the mechanism to separate synthetic polymer from natural polymers?

5. Why not hot needle test performed to confirm synthetic polymer?

6. Cause of contamination in decided sites are not properly address or concluded.

7. References are not properly arranged. kindly arrange alphabetically or year wise.

Reviewer 2 Report

This study is well-developed in terms of methodology and the approach and generally well presented with some interesting results.  Some minor questions/points are;

 Line 33: cloths: Change to clothes or clothing.

Line 103: Sampling net was thrawled just below surface and Line 254-258: Most abundant polymers were PET and semi-synthetic rayon. Can this be the result of density of the plastic polymer and availability of these near surface, as some plastic polymers have relatively higher density?

 Line 240: Microplastic contamination resulting from sampling method may constitute 12%. Is this a result of the control experiments mentioned in Line 120? Or is it a result obtained from the statictical analyses?

Line 152: Figure 2 demonstrates the microfiber/fragments found in bay areas as Line 165-166 states that there were no significant differences found with changing distance from the field station. Lakes on the other hand showed the distance and microfiber count correlation (Line 199-200). In discussion, lakes were mentioned in depth; yet, discussion about bay areas or potential points related to them are missing. Since the area is scarcely populated, it might be better to include some discussion for bay areas.   

 Line 368 and onwards (Conclusions). Some concrete results should be provided in this section as most readers would read the "conclusions" before looking at the main text.